# Is It Conjugated or Not? The Theoretical and Experimental Electron Density Map of Bonding in *p*-CH_3_CH_2_COC_6_H_4_-C≡C-C≡C-*p*-C_6_H_4_COCH_3_CH_2_

**DOI:** 10.3390/molecules25194388

**Published:** 2020-09-24

**Authors:** Przemysław Starynowicz, Sławomir Berski, Nurbey Gulia, Karolina Osowska, Tadeusz Lis, Sławomir Szafert

**Affiliations:** Department of Chemistry, University of Wrocław, 14 F. Joliot-Curie, 50-383 Wrocław, Poland; przemyslaw.starynowicz@chem.uni.wroc.pl (P.S.); slawomir.berski@chem.uni.wroc.pl (S.B.); nurbey.gulia@chem.uni.wroc.pl (N.G.); karolina.osowska@chem.uni.wroc.pl (K.O.); tadeusz.lis@chem.uni.wroc.pl (T.L.)

**Keywords:** polyynes, density map, single-crystal X-ray diffraction, theoretical calculations, electron localization function

## Abstract

The electron density of *p*-CH_3_CH_2_COC_6_H_4_-C≡CC≡C-*p*-C_6_H_4_COCH_3_CH_2_ has been investigated on the basis of single-crystal X-ray diffraction data collected to high resolution at 100 K and from theoretical calculations. An analysis of the X-ray data of the diyne showed interesting “liquidity” of electron distribution along the carbon chain compared to 1,2-diphenylacetylene. These findings are compatible with the results of topological analysis of Electron Localization Function (ELF), which has also revealed a larger (than expected) concentration of the electron density at the single bonds. Both methods indicate a clear π-type or “banana” character of a single bond and a significant distortion from the typical conjugated structure of the bonding in the diyne with a small contribution of cumulenic structures.

## 1. Introduction

Within the last three decades, research on organic, organometallic and metal-containing polyynes has focused a lot of attention and witnessed a series of stunning breakthroughs [1,2]. Such compounds of rigid-rod architecture are envisioned as precursors of still elusive sp-hybridized carbon allotrope—carbyne. Yet another important factor that significantly spurred the research in this area is the extraordinary (and somehow surprising) physical properties of such “material” that have been predicted by theorists. They suggest (based on theoretical calculations) that carbyne possesses for instance, a tensile strength that surpasses that of carbon nanotubes, graphene, and diamond [3,4,5].

In order to better understand the physical and chemical properties of polyynes and gain a deeper insight into the detailed electronic structure of a carbon chain, various series of oligo and polyynes of type X(C≡C)_n_X have been synthesized and thoroughly studied [6,7,8,9,10,11,12,13,14,15,16,17,18,19,20,21,22,23,24,25,26]. The received data allowed to define the effect of a chain length upon various molecular properties and quantify a dependence of many measurable physical values, like an absorption band, redox potential, or NMR chemical shifts against 1/*n* (where *n* is the number of alkyne units).

The structure of the polyyne chain itself is of great interest from several standpoints. One of the most intriguing questions is: With the chain lengthening, will the triple and single bond equalize (a vanishing HOMO/LUMO energy gap) or approach two different values (a persistent energy gap; from a solid-state physics perspective, a Peierls distortion) [27,28]? Another good question is: To what degree carbyne can easily bend? There has been conjecture that distortion of long sp carbon chains might trigger isomerization to fullerenes or other sp^2^ carbon allotropes [29,30,31,32,33,34,35].

In the 2003 review article (followed by the 2006 update paper), all crystallographically characterized compounds with at least eight consecutive sp hybridized carbons (four conjugated C≡C bonds) were analyzed [36]. The major interest—at the molecular level—were bond lengths, bond angles, and sp carbon chain conformations. Average values derived from bond length suggested that (1) as the midpoints of the sp carbon chains are approached, the C-C bonds contract and the C≡C bonds lengthen, and (2) as the chains are extended to the macromolecular limit of the one-dimensional carbon allotrope (carbyne), the C-C bonds contract to the proposed limiting value 1.32–1.33 Å and the C≡C bonds lengthen to 1.25 Å (the effect of the error limits on the bond lengths (esd values) is omitted).

Although in further approximations of these limits, computational chemists will still play a crucial role we believe some additional important experimental values can be gathered. This would of course, include precise crystallographic electron density maps of bonding in polyynes. Although, what is obvious, the main interest should be directed at longer polyynes as an initial point we present here the map of a diyne with a carbonyl group containing phenyl endgroups, which is next compared with a theoretical one.

## 2. Results and Discussion

The experimental difficulties of the research on electron density maps are worth mentioning. First, the highest quality crystals are required. Second, the compound and its crystals must be extremely stable and should not change (cracking, phase transitions, etc.) during an experiment. Third, there must not be any disorder in the measured crystal and data must be collected and processed with the utmost care. Long-lasting optimization of the crystallization process (speed, temperature, concentration) and pure luck allowed us to finally obtain crystals of sufficiently good quality to constitute a proper data source for such an experiment. The crystal structure was first solved conventionally and the results have already been published [37].

The diyne *p*-CH_3_CH_2_COC_6_H_4_-C≡CC≡C-*p*-C_6_H_4_COCH_3_CH_2_ (**1**) was synthesized as described earlier and it was fully characterized including ^1^H- and ^13^C-NMR and IR spectroscopy, MS spectrometry and X-ray crystallography [37]. Next, data for the carefully selected crystal were collected as described in the Experimental Section. The view of the structure is presented in Figure 1 and the metrical parameters from conventional (CR) and multipole (MP) refinement are compared in Table 1.

As it can easily be noticed, the inclusion of the multipole parameters has led to just subtle modifications of the bond lengths. The distribution of the electric charge is given in Table 2 and the multipole deformation density is presented in Figure 2a. The final residue density map as shown in Figure 2b was essentially featureless with *ρ*_res_ between −0.1 and 0.2 e/Å^3^.

The negative charge of the carbonyl O(1) atom of −0.59(5) is similar to those in strychnine (−0.49(7)) [38] and 3-acetylcoumarin (−0.51 and −0.58) [39]. The benzene ring as a whole is slightly positive, with the sum charge of 0.53. Similarly, there is also a deficit of electrons on the terminal ethyl group, with its total charge of 0.56. The anomalously high negative charge of H(52) (−0.28(9)) results probably from imperfect partitioning of the electron density between this atom and C(5). The same can be observed for the C(16)-H(16) pair. The excess of the electron density is accumulated on the two carbon atoms (within the symmetry independent half of the molecule) involved in the triple bonds—the sum of their monopole charges is −0.55.

The essential topological parameters of the charge distribution are summarized in Table 3.

The parameters for the triple bond C(1)-C(2), density in the bond critical point (*ρ*_cp_) and the value of Laplacian (Δ*ρ*_cp_), are somewhat larger than those reported for two crystallographically independent molecules of 1,2-diphenylacetylene (*ρ*_cp_ = 2.79 eÅ^−3^ for both, Δ*ρ*_cp_ = −21.0 and −20.8 eÅ^−5^) [40]. The values obtained for the present molecule are close to those for the ideal triple bond −2.82 eÅ^−3^ and −30.9 eÅ^−5^ [41].

Contrary to 1,2-diphenylacetylene molecules the ellipticity, *ε* (defined as λ_1_*/*λ_2_ − 1, where λ_1_ and λ_2_ are the negative values of the ∂^2^*ρ*_cp_/∂*x_i_*∂*x_j_* matrix) of the C(1)-C(2) bond is very low (0.02; 0.25 and 0.20 for 1,2-diphenylacetylene) and the perpendicular bond section is practically circular.

In the case of the aromatic ring *ρ*_cp_ with values, 2.12–2.21 eÅ^−3^ (av. 2.18 eÅ^−3^) resembles the one reported in 1,2-diphenylacetylene, whereas Δ*ρ*_cp_, in the range from −18.8 to −20.2 eÅ^−5^ (av. −19.5 eÅ^−5^) is slightly smaller than for the cited molecule (from −16.4 to −17.1 eÅ^−5^). The average ellipticity of the aromatic C-C bonds is 0.13, definitely smaller than for benzene molecule (0.23) and slightly smaller than in 1,2-diphenylacetylene [42]. The presented data indicate the charge migration from the aromatic rings to the triple bond, contrary to what was observed in the cited molecule.

The density in the critical point of the C(3)-O(1) carbonyl bond is similar to the density of carbonyl bond in other compounds (where this group is not a part of the peptide bond), e.g., 2.69(3) eÅ^−3^ in 1-(4-fluorophenyl)-3,6,6-trimethyl-2-phenyl-1,5,6,7-tetrahydro-4*H*-indol-4-one [43] or 2.856 eÅ^−3^ in estrone [44]. On the other hand the Laplacian value of −6.3 eÅ^−5^ is significantly higher than in these two compounds, where Δ*ρ*_cp_ was −24.38(7) and −17.56 eÅ^−5^, respectively. Rather surprisingly, the ellipticity of this bond of 0.04 is small as compared to what might be expected for a double bond. Nevertheless, this value is similar to the ellipticity of the carbonyl bond in estrone (0.05; for 1-(4-fluorophenyl)-3,6,6-trimethyl-2-phenyl-1,5,6,7-tetrahydro-4*H*-indol-4-one it was 0.28). As far as the single bonds C(5)-C(4), C(4)-C(3) and C(3)-C(14) are concerned, it may be noticed that *ρ*_cp_ mildly increases and Δ*ρ*_cp_ decreases. The latter is accompanied by moderate growth of ellipticity: 0.02, 0.04 and 0.08 for C(5)-C(4), C(4)-C(3) and C(3)-C(14), respectively. The parameters for the formally single bonds C(11)-C(1) and C(2)-C(2A) indicate that their character is close to aromatic, with the exception of small ellipticity of 0.05 and 0.04, respectively.

The simulation of the electronic properties of the crystal of **1** was performed next. Calculations for the experimental geometry yielded the lattice energy (E_cryst_), which strongly depends on the electron density functional chosen, as shown in Table 4. The E_cryst_ ranges from −100.7 kcal/mol (SVWN) to −28.2 kcal/mol (B3PW). After correction for BSSE effects in the counterpoise correction scheme [45] E_cryst_ has been reduced to −78.3 kcal/mol (SVWN) and −8.4 kcal/mol (B3PW). For further analysis, only the results achieved with the hybrid electron density functional B3LYP [46,47,48] were adopted (E_cryst_ = −54.9, E_cryst_ + BSSE = −32.3 kcal/mol), which are commonly used in studies on non-interacting molecules.

The electron density distribution calculated for the R-(C11)-C(1)-C(2)-C(2A)-C(1A)-(C11A)-R fragment (R = *p*-C_6_H_4_COC_2_H_5_) at the molecular plane is presented in Figure 3. One can distinguish a large concentration of the electron density in the region of C(1)-C(2) and C(1A)-C(2A) bonds (slightly smaller for the C(2)-C(2A)) and relatively small concentration in the area of the C(11)-C(1) and C(11A)-C(1A) bonds. The largest amount of electrons is observed outside the interatomic axis. Interestingly, the electron distribution in the C(1)-C(2) and C(1A)-C(2A) regions resembles a picture of two pairs of “banana” bonds, as previously suggested by Pauling for the double bond in ethylene [49]. Adopting the π-σ representation, one would postulate an essential π-type character for the C(1)-C(2), C(1A)-C(2A) and C(2)-C(2A) bonds. On this basis, it seems reasonable to represent the bonding in the carbon chain (simplified manner) with the R–C≡C–C≡C–R formula but Lewis structures containing the double C(2)=C(2A) bond should also be taken into account.

To study the nature of the chemical bonds in the C(11)-C(1)-C(2)-C(2A)-C(1A)-C(11A) atomic chain a topological analysis of the electron localization function (ELF) was carried out for the isolated molecule as outlined in the experimental section (theoretical calculations section). The ELF function was introduced by Becke and Edgecombe [50] as a “simple measure of electron localization in atomic and molecular systems”. Figure 4 presents the 3D plot of the ELF function with marked basin populations.

For each pair of atoms in the carbon chain, disynaptic bonding basins were observed: V(C11,C1), V(C1,C2), V(C2,C2A), V(C2A,C1A), V(C1A,C11A). The largest basins are found for the C(1)-C(2) and C(1A)-C(2A) bonds with a torus-like shape being in agreement with the observation of large electron concentration outside the interatomic axis. Each of the V(C1,C2), V(C2A,C1A) basins are associated with two attractors (local maxima) of ELF and it suggests multiple types of carbon-carbon bonding. In the case of the V(C11,C1), V(C1A,C11A) and V(C2,C2A) basins, only single attractors are localized, usually corresponding to single bonds. However, one has to remember that there is no clear correlation between a number of bonding attractors and bond multiplicity. As one could expect, the largest basin population ($\bar{N}$) of 5.17 e is computed for the V(C1,C2) and V(C2A,C1A) basins, which is very similar to 5.13 e calculated previously for the isolated HC≡CH molecule [51]. Such a large value of $\bar{N}$ characterizes the depleted triple bond and agrees with formal representation with R-C≡C-C≡C-R Lewis structure. The central C(2)-C(2A) bond having 2.74 e is more saturated by the electron density than the C(11-C1), C(11A)-C(1A) bonds with 2.46 e and from a topological point of view, it can be described as intermediate between double and single bonds. On this basis, an additional representation of the bonding may be proposed, which contains a cumulene chain of the carbon atoms. It is worth to notice, that shorter carbon-carbon bond corresponds to larger value of the basin population, for instance, r(C(11)-C(1)) > r(C(2)-C(2A)) > r(C(1)-C(2)) and respective basin populations: 2.46 e < 2.74 e < 5.13 e.

A “bridge” between the mathematical theory of the chemical bond contained in topological analysis of the ELF function (with non-integer values of the basin populations) and “classical view” where the chemical bond is formed through the sharing of electron pair(s) may be derived from a resonance of the Lewis structures. In Figure 5, five resonance forms are presented with corresponding percentages of occurrence in the mesomeric equilibrium.

The largest contribution has been computed for the structure I (54%), which corresponds to the R-C≡C-C≡C-R Lewis structure with an alternating triple (C(1)≡C(2), C(1A)≡C(2A)) and single (C(11)-C(1), C(2)-C(2A), C(1A)-C(11A)) bonds. In this structure, no formal charge is given to any atom (preferable form) and it reflects the large basin population (5.17 e) calculated for the V(C1,C2), V(C2A,C1A) basins. The Lewis structures II and III (symmetric forms) participate with 19% percentages to the equilibrium and reflect cumulene (^(+)^R=C=C=C=C^(−)^-R ↔ R-^(−)^C=C=C=C=R^(+)^) character of the bonding. Both resonance forms have formal charges −1, +1 at the C(1) and R and introduce a charge separation due to delocalization. Furthermore, they are responsible for the rigidity of the carbon chain and smaller than 6.0 e population between acetylenic carbons C(1) and C(2) but increase on the C(11)-C(1) and C(2)-C(2A) bonds. The Lewis structures IV and V (symmetric forms) represent the mixed cumulene-acetylenic character of the bonding. Both forms participate with a very small contribution of 4% to the resonance equilibrium. Comparing only contributions of the Lewis structures I to III, one can conclude that the bonding in the carbon chain possesses 1.42 times larger acetylenic than cumulene character. Furthermore, an approximate percentage ratio of three types of the Lewis structures (I:II, III:IV, V) may be simplified as 7:5:1.

## 3. Experimental Section

X-ray diffraction data were collected by using a KUMA KM4 CCD (*ω* scan technique, Kuma Diffraction, Wroclaw, Poland) diffractometer equipped with an Oxford Cryosystem cryostream cooler. Crystals of **1** suitable for an X-ray diffraction experiment were grown at room temperature (25 °C) from a CH_2_Cl_2_ solution gently layered on top with hexanes. The crystal selected for the measurement was picked directly from a mother liquor and immediately transferred to a diffractometer where it was cooled to 100 K.

### 3.1. Multipole Refinement

The multipole refinement was performed against 6073 merged reflections with *I* ≥ 3σ (*I*) with XD suite of programs [52], using the coordinates obtained from the conventional refinement. The C-H distances were fixed at 1.083 Å, in agreement with the typical values obtained from neutron diffraction data. The multipole formalism was described by Hansen and Coppens [53]. The multipoles were expanded up to octupoles for the C and O, and up to quadrupoles for H. The κ and κ’ parameters (separately for C and O atoms) were refined, whereas those for H were kept fixed at 1.2 (attempts to relax them failed). The topological analysis of the experimental charge density distribution was performed with XDPROP routine of the XD package following the ideas set out by Bader [54].

### 3.2. Theoretical Calculations

A simulation of electronic properties of the crystal has been performed for experimental geometry by means of the periodic ab initio DFT calculations using the CRYSTAL03 program [55]. Different combinations of the exchange and correlation functionals: SVWN, PWGGA, B3LYP and B3PW in conjunction with the 6-31G(d,p) basis set were employed [56,57]. A topological analysis of the electron localization function (ELF) [50] was carried out for the isolated molecule by means of the DGrid 4.2/Basin 4.2. programs [58,59]. Molecular orbitals have been generated for the crystal geometry at the B3LYP/6-311++G(d,p) computational level using Gaussian03 program [58,59,60,61,62]. The 3D plot of the ELF function was obtained by means of the Molekel program [63].

## 4. Conclusions

In conclusion, we have presented the first example of the experimental electron density map of the conjugated diyne *p*-CH_3_CH_2_COC_6_H_4_-C≡C-C≡C-*p*-C_6_H_4_COCH_3_CH_2_. The results clearly showed, that electrons in this arene-acetylene system are quite liquid. Compared to the results obtained for 1,2-diphenylacetylene, our data unambiguously indicated that insertion of an additional triple bond and concomitant conjugation in the -C≡C-C≡C- system remarkably change the charge distribution. This is due to the fact that conjugated triple bonds in **1** attract charge from the aromatic rings, contrary to the single -C≡C- bond in 1,2-diphenylacetylene. This experimental finding has been supported by theoretical studies on the chemical bonding in **1** performed with the use of topological analysis of the ELF function. Also, this method showed the bonding not to be a simply alternating single-triple-single-triple-single system, but revealed an enhancement of the electron density at the single bonds. This was to the extent that these bonds may be regarded as intermediates between the single and the double bonds.

## Figures and Tables

**Figure 1 molecules-25-04388-f001:**
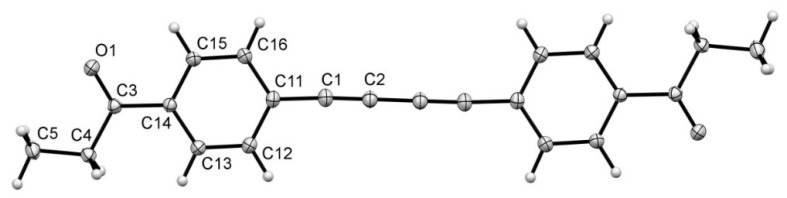
View of **1**.

**Figure 2 molecules-25-04388-f002:**
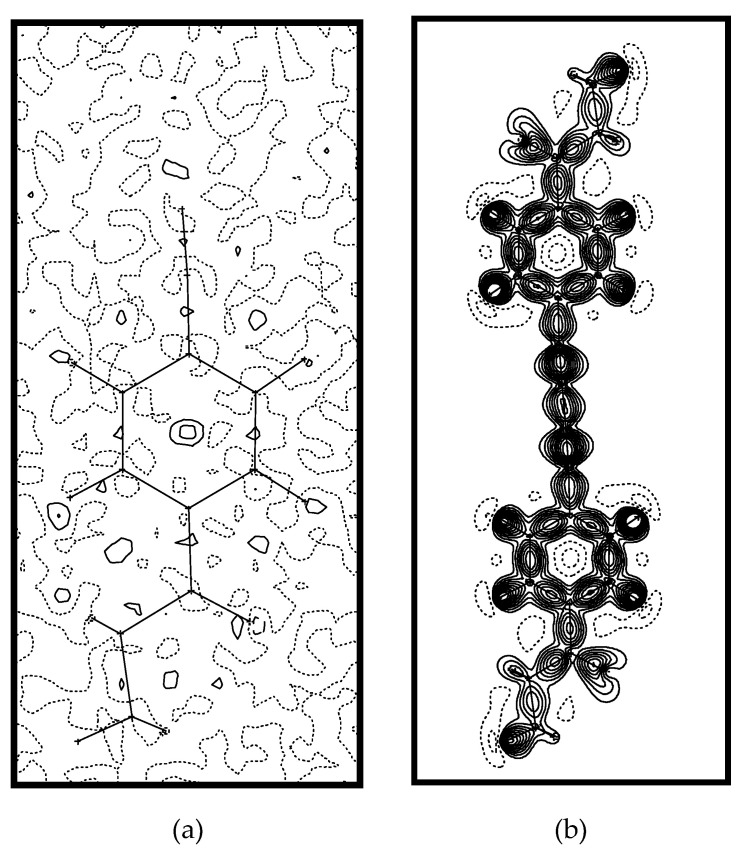
The final residue density (**a**) and the multipole deformation density (**b**) for **1**.

**Figure 3 molecules-25-04388-f003:**
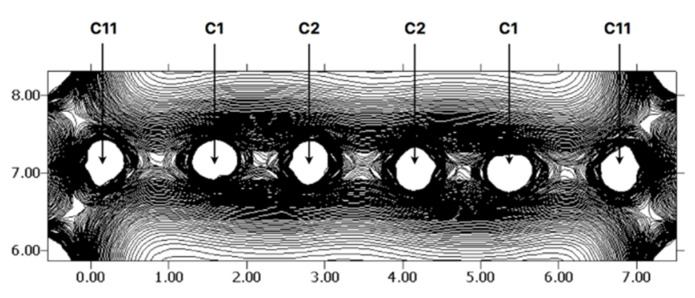
The electron density map *ρ*_bulk_ (r) of the R-(C11)-C(1)-C(2)-C(2A)-C(1A)-(C11A)-R fragment in the molecular plane. Lateral units in bohr and interval equal 0.005 e/bohr^3^.

**Figure 4 molecules-25-04388-f004:**
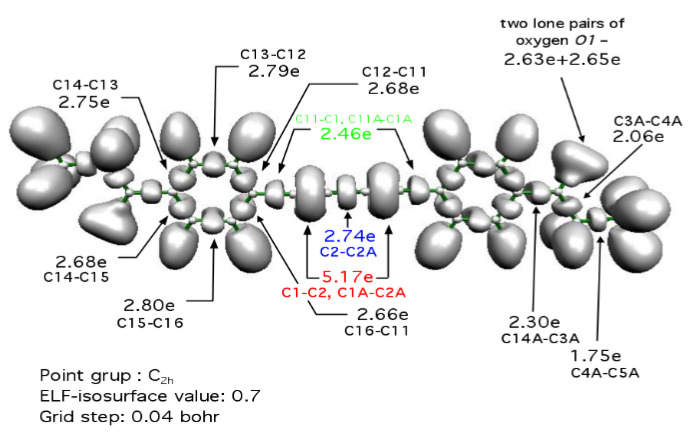
3D plot of the Electron Localization Function (ELF) for **1** calculated at the B3LYP/6-311++G(d,p) computational level. The bonding localization domains V(C,C), V(C,H), non-bonding domains V_1_(O)∪V_2_(O) of the oxygen atoms, and core domains C(C), C(O) are visible. The basin populations for V(C,C) and V_1_(O)∪V_2_(O) are presented in the picture.

**Figure 5 molecules-25-04388-f005:**
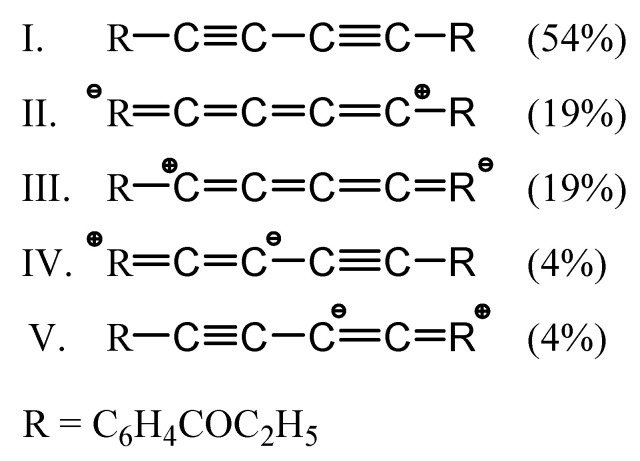
Percentage contributions of the Lewis structures representing the bonding in the R-(C11)-C(1)-C(2)-C(2A)-C(1A)-C(11A)-R carbon chain calculated using the basin populations ($\bar{N}$) obtained from the topological analysis of the ELF function.

**Table 1 molecules-25-04388-t001:** Bond lengths in **1**.

	CR	MP		CR	MP
O(1)-C(3)	1.2231(8)	1.2248(4)	C(11)-C(12)	1.4077(8)	1.4080(5)
C(1)-C(2)	1.2120(8)	1.2169(4)	C(11)-C(16)	1.4055(8)	1.4060(5)
C(1)-C(11)	1.4300(9)	1.4259(4)	C(12)-C(13)	1.3933(9)	1.3934(4)
C(2)-C(2A)	1.3670(11)	1.3636(6)	C(13)-C(14)	1.3981(8)	1.3987(4)
C(3)-C(4)	1.5141(8)	1.5109(4)	C(14)-C(15)	1.4039(8)	1.4039(4)
C(3)-C(14)	1.5016(9)	1.5008(4)	C(15)-C(16)	1.3908(9)	1.3924(4)
C(4)-C(5)	1.5202(9)	1.5237(5)			

**Table 2 molecules-25-04388-t002:** Pseudoatom charges for **1**.

Atom	Charge	Atom	Charge	Atom	Charge
O(1)	−0.59(5)	C(12)	0.10(4)	H(51)	0.12(9)
C(1)	−0.30(4)	C(13)	0.04(4)	H(52)	−0.28(9)
C(2)	−0.25(4)	C(14)	0.02(3)	H(53)	0.09(8)
C(3)	0.05(4)	C(15)	0.14(5)	H(12)	0.06(6)
C(4)	0.10(4)	C(16)	0.24(6)	H(13)	0.06(6)
C(5)	0.30(6)	H(41)	0.07(6)	H(15)	0.06(6)
C(11)	−0.05(3)	H(42)	0.16(5)	H(16)	−0.14(8)

**Table 3 molecules-25-04388-t003:** Topological parameters of the charge distribution in **1**.

Bond	*ρ* _cp_	Δ*ρ*_cp_	*R* _ij_	*d* _1_	*d* _2_	Hessian Eigenvalues			*ε*
						λ_1_	λ_2_	λ_3_	
O(1)-C(3)	2.78	−6.3	1.2249	0.7944	0.4305	−23.95	−22.93	40.59	0.04
C(1)-C(2)	2.92	−28.0	1.2170	0.6431	0.5739	−18.45	−18.14	8.54	0.02
C(1)-C(11)	2.01	−14.9	1.4260	0.7231	0.7029	−13.43	−12.81	11.33	0.05
C(2)-C(2′)	2.19	−15.3	1.3636	0.6818	0.6818	−13.95	−13.35	11.98	0.04
C(3)-C(4)	1.77	−12.6	1.5110	0.8174	0.6937	−11.47	−11.05	9.89	0.04
C(3)-C(14)	1.85	−14.8	1.5008	0.7521	0.7487	−12.81	−11.81	9.78	0.08
C(4)-C(5)	1.66	−10.2	1.5238	0.7655	0.7584	−10.22	−9.98	9.98	0.02
C(4)-H(41)	1.97	−19.7	1.0857	0.6240	0.4617	−16.78	−15.33	12.37	0.09
C(4)-H(42)	1.76	−15.2	1.0840	0.6834	0.4006	−15.43	−14.50	14.73	0.06
C(5)-H(51)	2.08	−22.7	1.0858	0.6276	0.4583	−17.76	−16.64	11.73	0.07
C(5)-H(52)	1.97	−15.5	1.0851	0.5402	0.5449	−15.04	−13.98	13.49	0.08
C(5)-H(53)	1.94	−19.1	1.0892	0.6296	0.4596	−16.48	−15.05	12.45	0.10
C(11)-C(12)	2.18	−19.5	1.4081	0.7208	0.6873	−15.57	−13.61	9.68	0.14
C(11)-C(16)	2.12	−19.0	1.4061	0.6881	0.7180	−15.36	−13.13	9.52	0.17
C(12)-C(13)	2.21	−19.5	1.3936	0.6888	0.7048	−15.48	−13.80	9.78	0.12
C(12)-H(12)	2.06	−21.8	1.0835	0.6186	0.4649	−17.36	−16.87	12.43	0.03
C(13)-C(14)	2.14	−18.8	1.3988	0.6811	0.7177	−14.92	−13.66	9.73	0.09
C(13)-H(13)	2.08	−22.7	1.0836	0.6383	0.4453	−18.12	−17.59	13.04	0.03
C(14)-C(15)	2.21	−20.2	1.4041	0.6904	0.7137	−15.73	−14.06	9.61	0.12
C(15)-C(16)	2.21	−20.1	1.3924	0.6783	0.7141	−15.81	−13.66	9.38	0.16
C(15)-H(15)	2.10	−23.4	1.0834	0.6068	0.4766	−17.76	−17.44	11.78	0.02
C(16)-H(16)	2.20	−23.4	1.0843	0.5232	0.5610	−18.18	−17.41	12.21	0.04

**Table 4 molecules-25-04388-t004:** Lattice energies (E_cryst_) before and after counterpoise correction calculated with different exchange and correlational functionals (all energies are in kcal/mol) using the 6-31G(d,p) basis set. The experimental geometry has been adopted.

Energy	SVWN	PWGGA	B3LYP	B3PW
E_cryst_	−100.7	−81.4	−54.9	−28.2
E_cryst_ + BSSE	−78.3	−58.1	−32.3	−8.4

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
