# Peer review of "Is It Conjugated or Not? The Theoretical and Experimental Electron Density Map of Bonding in p-CH3CH2COC6H4-C≡C-C≡C-p-C6H4COCH3CH2"

_molecules, 2020, doi:10.3390/molecules25194388_

Round 1
Reviewer 1 Report
Comments:
This work is analyzing the complex aspects of the conjugation - one of the central concepts in chemistry. In particularly, the conjugation between different systems could change the bonding scheme (perfect Lewis structure vs. biradical forms) and /or the contribution of polar forms vs. nonpolar ones. The case which was studied in this work is a conjugation between C≡C triple bond (diacetylene) and substituted phenyl group Ph (p-CH3CH2COC6H4-).
The electron density of Ph-C≡C-C≡C-Ph has been investigated on the basis of single crystal X-ray diffraction.
An analysis of the X-ray data of the 1,4-diphenyl-diacetylene showed the shift of electron population along the carbon chain compared to 1,2-diphenylacetylene. These findings are compatible with the results of a topological analysis ELF, which has also detected the shift of an electron density at the single bond.
Authors lead to conclusion that electrons in this arene-acetylene system are quite liquid. Authors declare this statement, but they don't propose the physical explanation. I would propose take into account the charge-shift effect and its connection with ELF - "Charge-Shift Bonding—A Class of Electron-Pair Bonds That Emerges from Valence Bond Theory and Is Supported by the Electron Localization Function Approach" of Shaik, Hiberty and Silvi, Chem. Eur. J. 2005, 11, 6358 – 6371.
I am also would propose compare the results with a very effective NBO analysis. NBO estimation of the Lewis structure contribution could strongly support the results of this work. NBO procedure is accessible in GAUSSIAN package.
In conclusion, the manuscript is publishable in Molecules, but should be reviewed again in revised form.
Author Response
First, we would like to thank for this suggestion. Generally, we agree with the reviewer. However, the mentioned analysis of the electronic structure, based on a number of Lewis structures in the framework of the Natural Bond Orbital Theory (NBO), may be professionally performed using the Natural Resonance Theory (NRT), proposed by Glendening and Weinhold. The NRT seems to be more suitable than NBO. Using the NBO methodology the single Lewis may be only considered. Unfortunately, the software which is able to perform such analysis (NRT) is out of our reach. It is also worthwhile to mention another argument against the use of the NBO (NRT). Both theories offer the approximation of the electronic structure in Hilbert space using the concept of the molecular orbital (MO). The molecular orbitals are not unique and may be defined in many ways. In our paper we proposed the topological approach to the electronic structure based on the ELF function, because the distribution of ELF in molecules is invariant with respect to certain types of unitary transformations among the reference molecular orbitals. Such representation of the electronic structure is much reliable than the approximation based on MOs.
Reviewer 2 Report
In this paper, the author analysed the bond length, the Electron Localization Function, the Lewis structure, and other characters of the conjugated diyne p-CH3CH2COC6H4-C≡C-C≡C-p-C6H4COCH3CH2. The methods used in this paper is adequately and the conclusions supported by the results. But the significance of the work is not clearly stated. The other question is that the result difference between SVWN, PWGGA, B3LYP and B3PW function are quite huge. What caused this difference and prompted the author adopt B3LYP。
Author Response
It is true that the lattice energy strongly depends on the exchange and correlation functionals used for calculations. The possible explanation of that result may be associated with the structure of the investigated crystal, which is stabilized, among others, by the C=O...H-C and C-H...π non-covalent interactions. Such interactions are weak and their proper description in the DFT theory strongly depends on the construction of a functional of the electron density. We have performed more calculations but they did not shed any extra light on the problem.
Reviewer 3 Report
This manuscript presents a study of the electron density of p-CH3CH2COC6H4-C=C-C=C-p- C6H4COCH2CH3 compound as a representative of the polyyne family. This family is very interesting and the study is well developed but, as the authors recognize in the introduction of the paper, the compound studied is a too short chain (only two conjugated triple bonds) to represent polyynes. On top of that, why was this particular compound chosen? Why this compound with Ph-COCH2CH3 groups at the end of the (short) chain? How is expected that these groups influence the properties of this polyyne?
The study includes experimental and computational points of view, but the results obtained are only mainly compared with 1,2-diphenylacetilene, what precludes the authors to derive significant conclusions. My recommendation is to enlarge the scope of the work, or to publish it in a different journal.
From the formal point of view, Figure 3 must be improved.
Author Response
- The crystal has been chosen since this was the only one of appropriate quality (see explanation below). On the other hand we were lucky that the crystal had ketone group since this is moderately electron withdrawing what influenced the carbon chain.
-
We fully agree that an enlargement of the scope of this work would be very valuable and informative. Nevertheless, this is something that cannot easily be done what we mentioned in the manuscript. The thing is that in order to get high quality and unambiguous experimental electron density map (based on x-ray) the crystal must be ‘perfect”. This means it has to be big enough and with basically no (even small) cracks, small in-growths, must be well-shaped, etc. and this is something that cannot be planned/predicted. We try to obtain such quality crystal with every polyyne that we synthesize in our lab, but we were lucky just in this single case so far.
- Figure 3 has been corrected.
Round 2
Reviewer 3 Report
Thank you to the authors for the sincere explanation of the reasons why their work is focused in this compound. I can understand the difficulty of obtaining a crystal of enough quality to performed this study, and I appreciate the work they have done. But the fact is that the reasons why I thought that this manuscript does not deserve publication in this journal are still valid: only one compound is studied, and it presents the minimum possible number of conjugated triple bonds (two), so the results obtained cannot be considered representative for the polyyne family, even more taking into account the presence of functional terminal groups.
On the other hand, the research design and the methods used to study this particular compound are adequate and the results and discussion are clearly presented, so I consider this manuscript merits to be published in another less general journal.
Author Response
Thank You very much again for the reviewer’s comments. I fully understand his/her point nevertheless I would like to one more time place some arguments to make our paper acceptable for publication.
For the last week I have searched the accessible databases to look at the result published using multipole refinement for different compounds. Actually, it appears that not so many (overall) papers (I have found a few dozen works) were published on the subject. Since the x-ray methods has been improved over the last years enormously and we can now measure extremely tiny crystals I figure that the problem must lie somewhere else. I do not believe that there is no interest in seeing real electron distribution over molecule since it is one of the fundamental points to observe where for instance electrophilic or nucleophilic “fragments” are located (to predict a molecule reactivity). This brings me to conclusion that the problem still lies in the quality of samples (extraordinary requirements for crystal quality to perform multipole refinement, what I have already mentioned before). I have never come across a systematic studies on more than one example of any family of compounds. Below I present few papers (but I have examined a lot more if not all of them) which prove that in all cases the measurement and refinement were performed on a single example.
- “Experimental and theoretical charge density, intermolecular interactions and electrostatic properties of metronidazole” Kalaiarasi, Chinnasamy; George, Christy; Gonnade, Rajesh G.; Hathwar, Venkatesha R.; Poomani, Kumaradhas Acta Crystallographica, Section B: Structural Science, Crystal Engineering and Materials (2019), 75(6), 942-953.
- “Topology of electron density and electrostatic potential of HIV reverse transcriptase inhibitor zidovudine from high resolution X-ray diffraction and charge density analysis” Iruthayaraj, Ancy; Chinnasamy, Kalaiarasi; Jha, Kunal Kumar; Munshi, Parthapratim; Pavan, Mysore S.; Kumaradhas, Poomani Journal of Molecular Structure (2019), 1180, 683-697.
- 3. „Investigations of the bond-selective response in a piezoelectric Li2SO4·H2O crystal to an applied external electric field” Schmidt, O.; Gorfman, S.; Bohaty, L.; Neumann, E.; Engelen, B.; Pietsch, U. Acta Crystallographica, Section A: Foundations of Crystallography (2009), 65(4), 267-275.
- „Role of the hydrogen bonds in nitroanilines' aggregation: Charge density study of m-nitroaniline” Pozzi, C. G.; Fantoni, A. C.; Punte, G.; Goeta, A. E. Chemical Physics (2009), 358(1-2), 68-74.
- „Studies on the electron density distribution of dolomite CaMg(CO3)2” Effenberger, H.; Kirfel, A.; Will, G. TMPM, Tschermaks Mineralogische und Petrographische Mitteilungen (1983), 31(1-2), 151-64.
- “Determination of Experimental Charge Density in Model Nickel Macrocycle: [3,11-Bis(methoxycarbonyl)-1,5,9,13-tetraazacyclohexadeca-1,3,9,11-tetraenato-(2-)-κ4N]nickel(II)” Domagala, Slawomir; Korybut-Daszkiewicz, Bohdan; Straver, Leo; Wozniak, Krzysztof Inorganic Chemistry (2009), 48(9), 4010-4020.
- “Investigation of Inter-Ion Interactions in N,N,N',N'-Tetramethylethylenediammonium Dithiocyanate via Experimental and Theoretical Charge Density Studies” Munshi, Parthapratim; Cameron, Elinor; Guru Row, Tayur N.; Ferrara, Joseph D.; Cameron, T. Stanley Journal of Physical Chemistry A (2007), 111(32), 7888-7897
- „An experimental charge density study of mesulergine hydrochloride, a dopamine agonist” Zhu, Naijue; Klein Stevens, Cheryl L.; Stevens, Edwin D. Journal of Chemical Crystallography (2005), 35(1), 13-22.
- „Role of the Hydrogen Bonds in Nitroanilines Aggregation: Charge Density Study of 2-Methyl-5-nitroaniline” Ellena, Javier; Goeta, Andres E.; Howard, Judith A. K.; Punte, Graciela Journal of Physical Chemistry A (2001), 105(38), 8696-8708.
- „Experimental electron density distribution of dopamine hydrochloride” Klein, Cheryl L. Structural Chemistry (1991), 2(5), 507-14.
Of course, I fully agree that the conclusions would be much more sound if the measurements were performed for let’s say a family of butadiynes with different end-groups or a family of polyynes with the same end-group but different carbon chain length. But this is basically very unlikely to see such job done. I can understand that a manuscript might not be accepted due to a narrow scope (of derivatives for instance), but in this case it is just a matter of pure luck and it would be unfair to “punish” us because we were not lucky. Knowing the literature on polyynes I still believe that the results are very interesting for the polyyne community. Many papers are dealing with electronic structure of polyyne chains using different measures (IR mainly) and in this case we for the first time present a real (experimental) electron density on a chain that is long enough to see some spectacular electron shifts. I truly hope that the above explanations will convince the referee to finally accept the paper. Thanks a lot for all the help and I promise that we will keep working to get example of electronic maps for longer polyynes (maybe even unsymmetrical).